TECHNICAL RELEASE

# RiboSnake – a user-friendly, robust, reproducible, multipurpose and documentation-extensive pipeline for 16S rRNA gene microbiome analysis

Ann-Kathrin Dörr[1], Josefa Welling[1], Adrian Dörr[1], Jule Gosch[1], Hannah Möhlen[1], Ricarda Schmithausen[1,2], Jan Kehrmann[3], Folker Meyer[1] and Ivana Kraiselburd[1,*]

1 Institute for Artificial Intelligence in Medicine, University Duisburg-Essen, 45131, Essen, Germany
2 Institute for Hygiene and Public Health, University Hospital Bonn, 53127, Bonn, Germany
3 Institute for Medical Microbiology, University Hospital Essen, 45147, Essen, Germany

## ABSTRACT

**Background:** Next-generation sequencing for microbial communities has become a standard technique. However, the computational analysis remains resource-intensive. With declining costs and growing adoption of sequencing-based methods in many fields, validated, fully automated, reproducible and flexible pipelines are increasingly essential in various scientific fields.

**Results:** We present RiboSnake, a validated, automated, reproducible QIIME2-based pipeline implemented in Snakemake for analysing *16S rRNA* gene amplicon sequencing data. RiboSnake includes pre-packaged validated parameter sets optimized for different sample types, from environmental samples to patient data. The configuration packages can be easily adapted and shared, requiring minimal user input.

**Conclusion:** RiboSnake is a new alternative for researchers employing *16S rRNA* gene amplicon sequencing and looking for a customizable and user-friendly pipeline for microbiome analyses with *in vitro* validated settings. By automating the analysis with validated parameters for diverse sample types, RiboSnake enhances existing methods significantly. The workflow repository can be found on GitHub (https://github.com/IKIM-Essen/RiboSnake).

**Submitted:** 29 May 2024

\* Corresponding author. E-mail: ivana.kraiselburd@uk-essen.de

Preprint submitted at https://doi.org/10.1101/2024.08.06.606757

**Subjects** Software and Workflows, Bioinformatics, Microbial Ecology

## INTRODUCTION

Computational processing of sequencing data has become essential in environmental and medical research, and it will continue to expand as new technologies emerge [1]. Importantly, computational methods are routinely applied by non-specialists as their adoption across many fields of science continues.

Next-generation sequencing technologies are widely employed for describing microbial communities associated with different environments, such as in the Human Microbiome Project [2] and the Earth Microbiome Project [3]. They are also applied in environmental microbial surveillance [4], as well as for unraveling the associations between the microbiome and human diseases [5–7].

In the medical context, rapid and reliable analysis of the composition of bacterial communities can be important for the early detection of infectious diseases. Within this area of application, the analysis of *16S rRNA* gene amplicon sequencing data remains a valuable tool for providing taxonomic information about microbial communities [8–11], as it is a low-cost and rapid turnaround method.

Analysis of the *16S rRNA* gene is performed by PCR amplifying one or several of nine hypervariable regions of the *16S rRNA* gene using primers targeting flanking conserved regions [9, 12]. Following next-generation sequencing of the amplified fragments, sequences are processed *in silico* to extract phylogenetic information about the microorganisms present in the original sample, as well as their abundance. In addition to filtering the data on the basis of sequence length and quality, key steps for *16S rRNA* gene analysis are sequence joining in case of paired-end sequences, clustering of sequences based on similarity [12] and aligning these clusters to a reference database to determine the corresponding taxonomy [13].

In both the environmental and medical contexts, the need for repeatability and the large sample size drives the demand for full automation and user-friendly approaches. This makes the use of a pipeline capable of processing large amounts of samples in parallel highly appealing.

The two most commonly used open-source software packages for *16S rRNA* data analysis are QIIME2 (RRID:SCR_021258) [14] and MOTHUR (RRID:SCR_011947) [15]. We chose QIIME2 as the basis for our workflow. Widely used and cited, it can be readily installed and offers a wide range of functionalities. However, analyses with QIIME2 require a significant number of parameters and users are frequently overwhelmed by the number of options at their disposal.

Ideally, a single set of analysis parameters, such as filtering thresholds or identity percentages for classification, would suffice for all analyses. However, different types of samples require fine-tuned parameters. We are convinced that *16S rRNA* analysis requires minimization of user interaction, standardization, and use of verified parameter sets. This is particularly important for the reproducibility of scientific results [16, 17] and to simplify the analysis for non-experts.

To accomplish this goal, three components are required: (1) a fully automated and reproducible pipeline that can be easily installed and maintained, (2) an easy-to-navigate output accessible to non-bioinformatic experts and (3) sets of validated parameters for different types of samples.

To establish automated processing, we preferred a workflow system over a custom shell script or program to leverage existing workflow systems and avoid re-implementing established concepts. We chose the Snakemake workflow system (RRID:SCR_003475) [18] for our pipeline.

Several pipelines for *16S rRNA* analysis have been previously published. Some of these use Snakemake with or without QIIME2, such as Tourmaline (RRID:SCR_022465) [19], Natrix [20] and others [21, 22] (Table 1). Others do not use Snakemake but rather bash or R-scripts, like EasyAmplicon [23].

While these methods facilitate *16S rRNA* analysis with minimal user interaction and yield results that include the taxonomic classification of bacteria in the microbial community, none of the previously mentioned pipelines includes the additional steps necessary to obtain the full benefits of the analysis. Furthermore, the graphical outputs are not readily



```
git clone https://github.com/IKIM-Essen/RiboSnake.git
snakemake --cores all --use-conda data_prep
snakemake --cores all --use-conda
```

**Figure 1.** The installation and execution of RiboSnake is straightforward and has minimal complexity, enabling a wide range of users.

**Table 1.** Our comparison of features of selected published pipelines automating *16S rRNA* community analysis shows the need for a novel system that provides an all-in-one pipeline, providing the user with a variety of results. This includes diversity and feature importance metrics, as well as default parameters tested on MOCK communities. In addition, the parameters were tested on human and environmental samples.

| Pipeline | Fully automated | Support for parameter setting | Sequence representation | Diversity analysis | Feature Importance analysis | Main software | Validated default parameters on MOCK |
|---|---|---|---|---|---|---|---|
| Natrix [20] | Yes | Yes | OTU or ASV | No | No | DADA2, Swarm | No |
| Tourmaline [19] | No | Yes | OTU | Yes | No | QIIME2 | No |
| Cascabel [21] | Yes | Yes | OTU or ASV | No | No | QIIME, MOTHUR, DADA2 | Yes |
| Dadasnake [22] | Yes | Yes | OTU or ASV | No | No | DADA2 | Yes |
| RiboSnake | Yes | Yes | OTU or ASV | Yes | Yes | QIIME2 | Yes |

accessible for most of the mentioned pipelines, and most importantly, the parameter settings are left to the end-user, which may lead to inconsistencies in the analysis.

We present RiboSnake, an open-source pipeline published under an MIT license, built for *16S rRNA* analysis using QIIME2 and wrapped in a Snakemake workflow. The pipeline minimizes the required user interaction for the installation and execution to three commands (Figure 1), assuming Conda (RRID:SCR_018317) and Snakemake are already installed. The execution requires only two user inputs; after the second input, the complete pipeline runs automatically. Additionally, we provide a set of *in vitro* validated parameters for different use cases. However, we allow full flexibility as users can provide additional parameter sets, using parameters that are either validated or defined *ad hoc* to meet their specific needs. Thus, we retain repeatability and ease of use for end users.

To provide suitable parameters for a number of different scenarios, we chose to re-sequence existing samples after spiking them with a MOCK community [24] to test the recovery of said MOCK against the background of different potentially complex matrices (e.g., soil or human blood).

The *16S rRNA* gene analysis performed with RiboSnake provides information in addition to the taxonomic classification and relative abundances of microorganisms, such as alpha- and beta-diversity analyses. Additional features include feature importance evaluation, and longitudinal analysis in case of time-dependent data. All the results are presented in one structured report, including the provenance information and parameters used for each step of the analysis. This approach ensures easy tracking and reference of the parameters used even months after the analysis.

The RiboSnake pipeline offers maximum flexibility with a wide range of tools and analysis options that can be conveniently set via a config file. The documentation providing information on how to change the parameters can be found in the pipeline's GitHub repository. While individuals with limited training in *16S rRNA* analysis can choose from one of the existing default parameter sets that are distributed with RiboSnake, we assume expert users will set all parameters according to their needs and thus create novel

parameter sets. We also enable the sharing of new sets by including user-contributed parameters with and without *in vitro* verification. Specific parameter sets can be submitted via a pull request to the GitHub repository into the contributions directory.

## REQUIREMENTS AND DESIGN

### Requirements

Different platforms, sequencing technologies, sequencing chemistries and even primer locations play a significant role. Accordingly, the implementation needs to be adaptable to accommodate various input data types. Repeatability is another key issue, as is minimizing the need for user-generated documentation and user input. These issues are interrelated, and providing profiles that aggregate parameters serves both purposes. In addition, end-user-friendly visualization of results is needed. Finally, the pipeline needs to provide both longevity and portability.

Ease of installation and maintenance of the deployed instances are additional goals of our project.

### Design

By using the well-established and permanently maintained software platforms Conda (RRID:SCR_018317) [25], QIIME2 [14] and Snakemake [18], we leverage existing community investments and achieve longevity and multi-platform support.

Snakemake and Conda ensure the presence of all necessary software packages and their corresponding versions, providing the required execution environment. Snakemake's best practices [26] were followed when creating this pipeline. The Snakemake report consolidates all analysis results, including tables and visualizations, into a single document. Intermediate results that could be interesting for special analysis steps can be found in the general result's folder.

Validated, appropriately named parameter sets were created and made available, ensuring that RiboSnake users chose the best parameter combinations for each sample matrix.

## DATA AND METHODS

Sequence data for parameter set validation was generated using an Illumina Miseq sequencing instrument and the V2 chemistry. An overview of all processed sample types can be found in Table 2.

### Clinical data

As proxies for clinical data, skin and work environment surfaces were sampled by swabbing the hands of three individuals and their respective keyboards using cotton swabs pre-moistened with TE buffer +0.5% Tween 20. The swabs were stored at −80 °C until DNA extraction with the DNAeasy PowerSoil DNA Isolation Kit (Qiagen), according to the manufacturer's instructions. Plasma samples were obtained by centrifugation of blood taken from non-septic patients, followed by DNA extraction using Qiagen's QIAamp UCP Pathogen Mini Kit. Samples taken from rectal swabs, published in [27], were analysed as a third kind of human-associated samples. The authors described a similar sampling process and storage to the hand swabs, with the exception of using the ZymoBIOMICS DNA Miniprep Kit (Zymo Research) for DNA extraction.

**Table 2.** Parameter sets created in this research, including the sample types, DNA extraction kits used, library preparation, and validation methods.

| Name | Sample origin | ID | Extraction kit | Library preparation | Validation |
|---|---|---|---|---|---|
| QIAamp-Illumina-human-blood | Human blood | PRJEB75965 | Qiagen's QIAamp UCP Pathogen Mini Kit | 16S Metagenomic Sequencing Library Preparation (Illumina) | MOCK spike-in |
| PowerSoil-Illumina-human-skin | Human skin | PRJEB75965 | DNAeasy PowerSoil DNA Isolation Kit (Qiagen) | 16S Metagenomic Sequencing Library Preparation (Illumina) | None |
| ZymoBIOMOCS-Illumina-human-gut | Human gut | PRJNA747262 | ZymoBIOMICS DNA Miniprep Kit | NEBNext® Ultra II DNA Library Prep Kit | Published research |
| PowerSoil-Illumina-work-surface | Surface | PRJEB75965 | DNAeasy PowerSoil DNA Isolation Kit (Qiagen) | 16S Metagenomic Sequencing Library Preparation (Illumina) | MOCK spike-in |
| innuPrep-Illumina-wastewater | Wastewater | PRJEB75965 | innuPrep AniPath DNA/RNA | | MOCK spike-in |
| PowerSoil-Illumina-soil | Soil | 342 (MG-RAST) | PowerSoil®-htp 96 Well Soil DNA Isolation Kit | | Published research |
| FastDNA-Illumina-sand | Sand | ERP019482 (EMBL/EBI) | FastDNA spin kit for soils (MP Biomedicals, LLC, Solon, OH) | | Published research |

## Environmental data

A number of environmental samples already exist in the literature (e.g., the Atacama desert [28] and grasslands [29]). While we can use existing parameter sets, other areas (e.g., wastewater) lack verified parameter sets. We obtained wastewater samples from different sewers located in the city of Essen (Germany), as well as a regional wastewater treatment plant. The sampling procedure and locations are described in [30]. Aliquots of 200 mL of the wastewater samples were filtered using electronegative filters with 0.45 µm pore size (MF-Millipore), followed by nucleic acid extraction using the innuPREP AniPath DNA/RNA Kit on an InnuPure C16 touch device (Analytik Jena).

## MOCK community spiking and recovery

To identify adequate analysis parameters, we used a commercial MOCK community using intact cells (ZymoBIOMICS™ Microbial Community Standard Catalog No. D6300). The cells were spiked in the respective samples and subjected to DNA extraction according to the methods described above. We used the DNA (ZymoBIOMICS™ Microbial Community DNA Standard Catalog Nos. D6305) as positive controls for amplicon PCR. Subsequently, we identified the optimal parameters to recover the MOCK community signal in the respective sample types.

## Library preparation and sequencing

For all samples processed in our laboratory, sequencing libraries were prepared from the extracted DNA according to the 16S Metagenomic sequencing Library Preparation protocol (Illumina), using the primers Bakt_341F and Bakt_805R targeting the V3-V4 region [31]. Libraries were sequenced on an Illumina MiSeq instrument using the 2×250 V2 chemistry.

## FEATURES AND IMPLEMENTATIONS

RiboSnake combines several logical steps to provide a complete *16S rRNA* analysis, starting with quality control, and removal of technical sequencing artifacts and contaminations. Subsequently, sequences are clustered based on similarity and a taxonomy is assigned to the cluster representatives from a reference database. A variety of statistical methods are employed in the computation of diversity metrics and feature importance analysis. Visualizations of the results, as well as summaries of the parameters and quality metrics, are included in the final report.

For preprocessing, quality control, information about per base quality, read counts and duplications is obtained with FastQC (RRID:SCR_014583) [32]. In combination with Kraken2 [33] information, which provides a first taxonomic overview, both results are visualized with MultiQC (RRID:SCR_014982) [34]. Trimming based on quality scores and sequence length, as well as clipping of adapters and primers, is performed with cutadapt (RRID:SCR_011841) [35]. Reads and regions showing a low-quality score are removed with the QIIME2's q-score method. Paired-end sequences are joined using Vsearch (RRID:SCR_024494) [36].

Depending on the type of sample, sequences obtained might include contaminations, such as human origin. These can be filtered out with the QIIME2 implementation of BLAST [37]. As an alternative to BLAST, which provides results more rapidly and is not utilized with its QIIME2 implementation, bowtie2 (RRID:SCR_016368) [38] has been implemented. As contamination with undesired DNA sequences can vary between samples, filtration parameters should be adjusted accordingly. The filtering parameters for the sample at hand can be set based on preceding quality inspection or information from other research dealing with the same kind of data. By changing the reference sequence in the config file, contaminations from different origins can be eliminated. One example would be the analysis of stool samples from humans or from mice, where the analysis would require filtering out the different host genomes as possible contaminants.

For filtering out chimeric sequences, *de novo* clustering into operational taxonomic units (OTUs) [39] and classification, Vsearch [36] is used. To increase the robustness of the analysis and reduce false positives [40], abundance filtering is performed after classification. Default values for the abundance threshold were chosen according to Nearing *et al.* [40]. The efficacy of this methodology entails the exclusion of reads based on frequency and prevalence, a process analogous to that employed by tools utilized for decontamination, such as decontam [41]. The setting of this filter depends on the sample type as well as the research question at hand. We recommend lowering the filter in case of a large number of samples or when specifically looking for low-abundant bacteria.

The user can switch easily between using amplicon sequence variants (ASVs) [42] and OTUs through settings in the config file. For ASV generation, the pipeline uses DADA2 (RRID:SCR_023519), a package for correcting Illumina sequencing errors and clustering sequences [43].

Different databases for taxonomic classification are supported and can be specified in the config file. The default is the SILVA database [44]. Greengenes2 [45] can be used alternatively. It is also possible to create a database from NCBI data with the help of the QIIME2 plugin of RESCRIPt [46]. When specified, the references are downloaded automatically. For each listed option, the results of the analysis are evaluated to ensure that the bacterial genera identified are consistent. In addition to describing the taxonomic composition of the samples, various diversity metrics can be computed based on the taxonomic classification. The within-sample diversity, also called alpha-diversity, is a measurement of bacterial richness and evenness. Beta-diversity, on the other hand, calculates the diversity difference between two samples, giving information about their similarity [39]. Before the diversity analysis, a repeated rarefaction step is performed, as proposed by Cameron *et al.* [47]. This ensures that any potential bias in the analysis due to differences in sequencing depth between samples is eliminated [48]. Diversity calculations

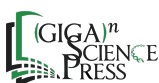

are performed with and without taking the phylogenetic distance of the observed bacteria into account.

A wide range of diversity analysis parameters are available in QIIME2 and its implemented diversity tools. These include the Shannon diversity index to measure richness, the Pielou index to measure evenness, and the Pearson or Spearman indices to perform standard correlation analyses. Analyses can be performed for several different parameters at once, and we include these options in the config file (commented out) for ease of adoption by end users.

For further in-depth analysis, feature importance analyses are performed using ANCOM [49], gneiss [50] and songbird [51]. Such analyses identify which bacterial genera are important or change the most with respect to a metadata property. While the songbird output is visualized with qurro [52], ANCOM and gneiss outputs are visualized with QIIME2 visualizations. In case the samples belong to a time series, the QIIME2 longitudinal plugin supports the analysis of volatility or feature-volatility over a continuous variable like time or temperature. Finally, a Snakemake report is created automatically, holding all essential information concerning filtering metrics, taxonomic classification and diversity analysis results. All report files have descriptions added that can help understand the results. The taxa barplot, the visualizations of alpha- and beta-diversity results, longitudinal analyses, filtering and quality information are added to the report file with all visualization and interaction options provided by the QIIME2 view. Further plots and the outputs of other tools like MultiQC are either added as html- or png-files. All these visualizations can be found in the different categories of the report file. Upon opening the RiboSnake report, users can access the different graphics and customize the colors based on the possibilities provided by the visualization package QIIME2.

With the exception of songbird, qurro and bowtie2, all tools included in RiboSnake were set up with their QIIME2 implementation. The required environments for the tools to run are created by Snakemake using Conda [25], ensuring that the necessary dependencies for all tools are present in the specific environment.

For running RiboSnake, the only installations that need to be done manually are those of Conda and Snakemake. The installation of all other packages is managed automatically.

The pipeline assumes the input consists of a set of FASTQ files (we currently support either single-end or paired-end Illumina data) as well as a text file containing metadata in a specific format, and the pipeline subsequently converts this to a QIIME2-supported metadata format. The pipeline currently does not support data processed with sequencing platforms other than Illumina, like Oxford Nanopore or Thermofisher. The performance has also not been tested for full-length *16S rRNA* gene sequencing results.

When only taxonomic information is needed as output, an abbreviated analysis can be run omitting all diversity and feature importance analysis steps. All parameters can be set according to the user's needs in the workflow config file.

We note that we set the default for the different parameter sets according to the pipeline results for the bacterial abundances in samples, standardized by adding MOCK community cells.

The workflow was created following the best practices for creating a Snakemake workflow [26] and has been released on GitHub [53] under an MIT license. The workflow is embedded in a continuous integration and continuous delivery environment. This type of production environment applies ongoing automation, testing and monitoring to improve

the development process, resulting in reliable and functioning software capable of generating reproducible and credible data.

All tools used for this workflow can be found in Table 7 of the Appendix.

## USAGE AND FINDINGS

RiboSnake has been tested on several different sample types and configurations to ensure its usability for a wide variety of applications. The details on installing and configuring RiboSnake are located on GitHub [53]; also, see Figure 1. To set the default parameters, the performance was first tested on samples spiked with MOCK community cells. With the resulting parameter sets, the pipeline was run for human and environmental samples, either paired-end or single-end sequencing reads. The parameters were set for ASVs as well as for OTUs. While most of the analyzed data correspond to amplicon sequencing of the V3-V4 region, a subset of the processed samples was obtained with primers proposed in the Earth Microbiome Project directed to the V4 region [54, 55].

### MOCK community tests for parameter optimization

In order to identify the best standard parameter sets for the pipeline for different sample types and to ensure their correctness, samples spiked with a MOCK community were used as standardized control [24]. The default parameters were chosen for the pipeline to retrieve all eight bacterial genera reported to be present in the MOCK community. Figure 2 shows that, upon processing the MOCK community as stand-alone DNA or cells, the pipeline reports the relative abundances of the eight bacteria at a similar abundance to that reported by the manufacturer. To note, the manufacturer points out that different extraction procedures and processing methods may result in changes in the bacterial abundance obtained (ZymoBIOMICS™ Microbial Community Standard Catalog No. D6300), as can be seen in Figure 2d where the relative abundances do not correspond exactly to the theoretical abundances. Furthermore, the relative abundance reported by RiboSnake pertains to the genus level, whereas ZymoBIOMICS reports abundance at the species level. This may account for the slight discrepancies observed between the relative abundances reported by RiboSnake and ZymoBIOMICS shown in Figures 2a to 2d. The abundances of the retrieved bacteria fluctuate more significantly as the complexity of the sampling matrix increases, as in Figures 2a to 2c. In Figure 2a, the sampling matrix is a sterile surface swab, while in Figure 2c, it is a plasma sample from a non-septic person. Both matrices are rather uncomplicated, as little to no bacterial genera are expected to be found in either of them. Opposite to that, the sampling matrix wastewater, depicted in Figure 2b, is expected to hold a high number of different bacterial genera. This makes it more difficult to retrieve the MOCK community, as the added genera become a minority to the existing ones.

The exact configuration parameters employed for retrieving the expected MOCK community bacteria with each type of sample can be found in Tables 5 and 6 of the Appendix.

### Diversity analysis

Table 3 shows the comparison of the expected and obtained diversity in different sample types. Using the verified parameter sets, we obtained diversity values that were in line with our expectations.

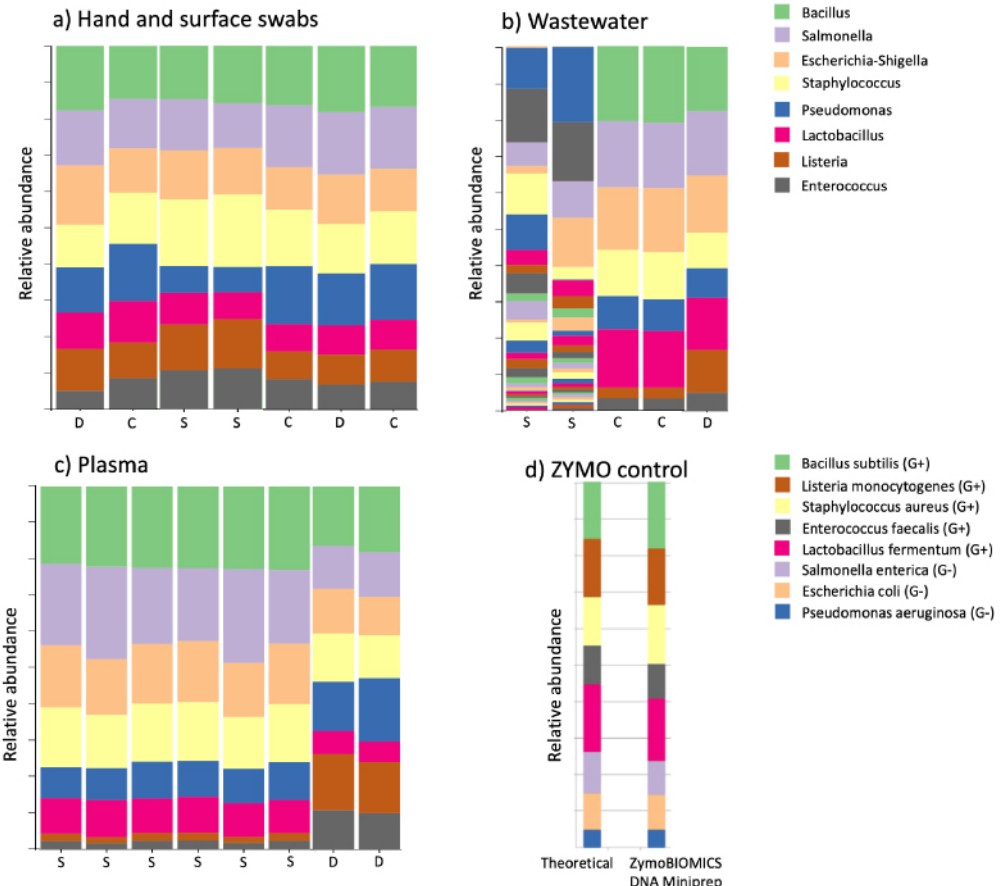

**Figure 2.** Taxonomy plot showing that after the analysis with RiboSnake, the eight expected genera from the MOCK community could be retrieved nearly every time, independently of sequencing run and sequenced samples. The same MOCK community, cell, or DNA was processed and sequenced several times with different additional samples. The DNA of the bacterial communities of every run was amplified with the primers for the V3-4 region. The bacteria in the MOCK community were retrieved for every sequencing run with frequency variations for some samples. In some sequencing runs, more than one positive control with a MOCK community was included. The communities were used as pure DNA (D), cells (C), or as spike-in (S). In panel d, the analysis results provided by ZymoBIOMOICS were added for comparison. The panel shows both the theoretical composition and the composition after analyzing with the ZymoBIOMICS DNA Miniprep kit. The theoretical composition was calculated by the manufacturer based on the theoretical genomic DNA composition.

**Table 3.** Expected and obtained diversity added to compare the general results for consistency. The obtained diversity is the mean value of the Shannon entropy for all samples analyzed of the same type.

| | Human | | | Surface | Environment | |
|---|---|---|---|---|---|---|
| | **Skin** | **Rectal swab** | **Plasma** | **Keyboard** | **Wastewater** | **Soil** |
| Expected Diversity | medium | medium | low | medium | high | high |
| Obtained Diversity | medium (mean: 3.06) | high (mean: 5.9)* | low | medium (mean: 3.52) | high (mean: 6.57) | high (mean: 5.51) |

*While generally expecting a medium alpha-diversity for this kind of sample, the shown results align with the results from the original study employing this data set [27].

For gut microbiome samples, the data available from Reinold *et al.* [27] were employed. When comparing their results with the RiboSnake run, slight discrepancies can be seen in the diversity metrics. While the calculated Shannon entropy is a little lower (RiboSnake: 5.9, reported: 6.5), the Pielou evenness is slightly higher (RiboSnake: 0.88, reported: 0.71).



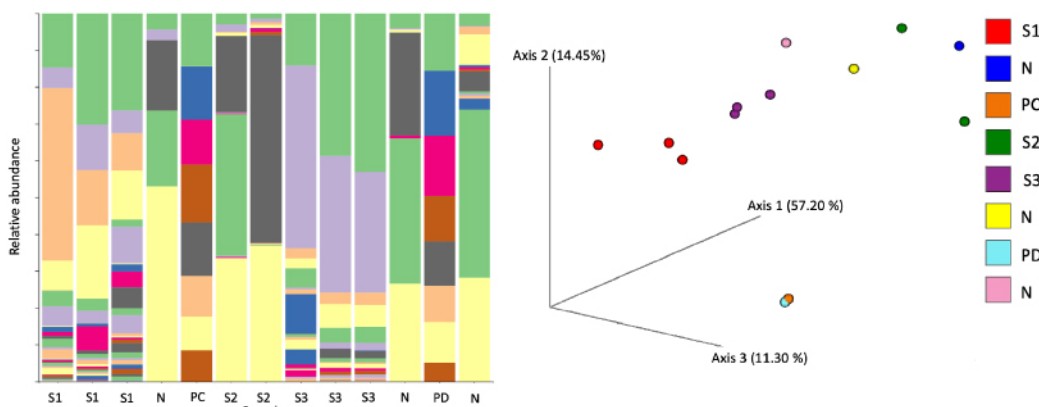

**Figure 3.** The plot shows two of the many outputs of RiboSnake. On the left, the relative frequency of the different bacterial genera per sample is depicted as a bar plot. We can see that the positive controls (PC, PD) show all eight expected bacterial genera. Each color depicts one bacterial genus; the corresponding legend can be found in Figure 7 of the Appendix. On the right, the results of the principal coordinate analysis are depicted to visualize the similarity of the different samples. The plot is an EMPeror plot based on an Euclidean beta-diversity analysis [56]. Samples taken from the same subjects cluster together based on their community composition. PCoA is a dimensionality reduction technique that calculates the distance between data points [56]. The closer the data points, the greater the similarity. S1 to S3 are the samples taken from subjects 1 to 3; the negative controls are marked as N.

The differences can be explained by the usage of OTUs instead of ASVs for the analysis, as fewer unique reads are conserved with OTUs.

## Human and surface swab samples

Figure 3 shows the taxonomic classification of the features found in the samples from hand and surface swabs. We recovered all eight bacterial genera of the MOCK community and were able to cluster the samples taken from each person's hands and keyboard together based on the beta-diversity results. Visualization of the clusters was performed with an EMPeror plot (RRID:SCR_024013) [56], showing the results of a principal coordinate analysis (PCoA).

The EMPeror plot in Figure 3, based on an Euclidean beta diversity metric, shows that the positive controls, as well as the samples taken from subjects 1 and 3 (S1 and S3), cluster together. This shows that the samples taken from one subject but different sampling sites display quite similar beta diversity. Since all samples from the same person clustered together, these results seem plausible [57].

These results show that the pipeline is capable of performing analysis steps beyond taxonomic classification with well-defined parameters. Both the analysis based on OTUs and the analysis based on ASVs produce equally satisfactory results, as shown in Figure 4. Differences can be seen in the number of bacterial genera found in the negative controls and the samples taken from subject 2 (S2). For nearly all samples, the analysis based on ASVs resulted in a higher number of bacterial genera. At the same time, the taxa with the highest abundance were found using both methods. This difference may have been due to variations in how single bases and error thresholds were handled, depending on the usage of ASVs or OTUs and their corresponding tools. While similar sequences were clustered to create OTUs, error thresholds for individual mutations were employed to create ASVs in order to obtain a number of unique sequences with high confidence [58, 59]. It is worth

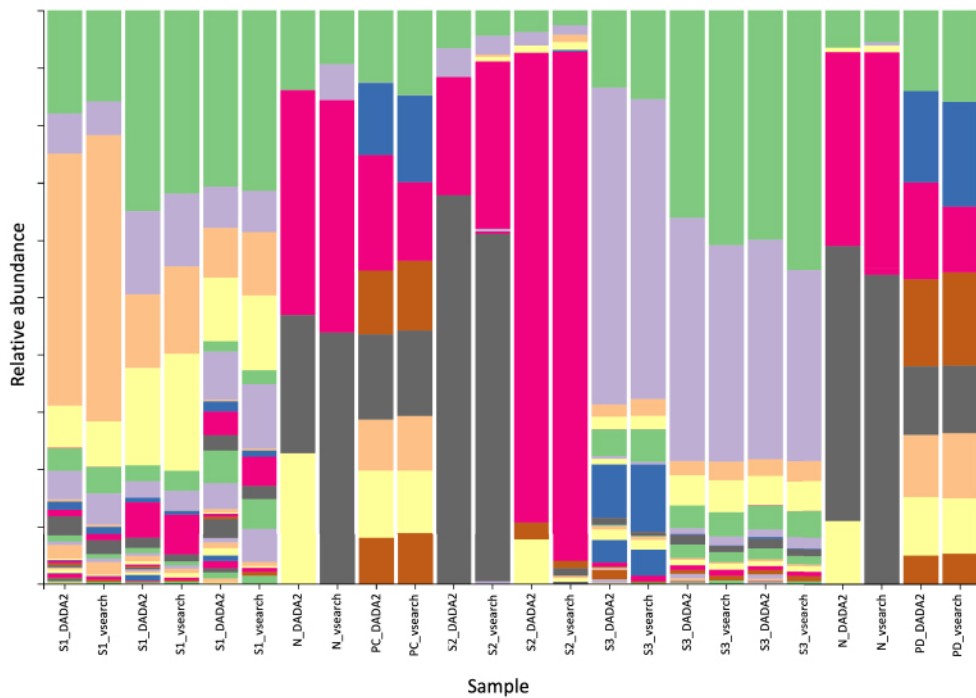

**Figure 4.** As the pipeline is capable of working with OTUs (vsearch) as well as with ASVs (DADA2), a comparison of the results using these methods on the same data is shown here. The results for the positive controls (PC, PD), as well as for subjects 1 (S1) and 3 (S3), match closely, while the results for the negative controls (N) and subject 2 (S2) vary. Each color depicts one bacterial genus; the legend can be found in Figure 8 of the Appendix.

mentioning that the absolute abundances of the OTUs in the negative controls were significantly smaller than for the rest of the samples (Figures 5 and 6 in the Appendix). Generally, getting classification results for the negative controls does not need to be a sign of contamination and could be caused by multiplexing artifacts [60].

## Environmental samples: wastewater and soil

For testing the pipeline on data generated from complex environmental samples, sequences obtained from wastewater and soil samples were processed. For wastewater samples, data revealed a more rapid decline in per-base quality than for human samples. This likely resulted from the high concentration of inhibitors remaining after the extraction process. Notably, running the environmental data with the parameters optimized for human samples made it impossible to retrieve all the bacterial genera present in the stand-alone MOCK community. Therefore, the filtering threshold for this analysis was lowered to retrieve features with a small relative abundance. The differences in the used parameters can be found in Tables 5 and 6 in the Appendix. This example shows vividly why different parameter sets are needed depending on the processed data. The sample type itself and the difference in sample processing, such as the DNA extraction method, could require parameter adjustments.

To check the usability of RiboSnake for data collected from soil samples, we used the Atacama soil dataset published in 2017, with samples taken from the Atacama desert [28], and samples collected from soil under switchgrass published in O'Brien *et al.* [29]. As the V4

**Table 4.** The ten most abundant phyla (in the case of *Proteobacteria* classes) from the original study by O'Brien *et al.* [29] and the re-analysis of 8 samples by RiboSnake. Differences in the naming conversions result from the usage of different databases (Greengenes and SILVA) as well as a time-dependent difference.

| Bacterial phyla/class O'Brien *et al.* [29] | Relative abundance | Bacterial phyla/class RiboSnake | Relative abundance |
|---|---|---|---|
| *Verrucomicrobia* | 0.316 | *Verrucomicrobiota* | 0.394 |
| *Acidobacteria* | 0.153 | *Acidobacteriota* | 0.189 |
| *Deltaproteobacteria* | 0.088 | *Gammaproteobacteria* | 0.108 |
| *Bacteroidetes* | 0.077 | *Myxococcota (Deltaproteobacteria)* | 0.065 |
| *Alphaproteobacteria* | 0.067 | *Bacteroidota* | 0.062 |
| *Betaproteobacteria* | 0.057 | *Alphaproteobacteria* | 0.059 |
| *Gammaproteobacteria* | 0.040 | *Desulfobacterota* | 0.017 |
| *Actinobacteria* | 0.038 | *Nitrospirota* | 0.011 |
| *Planctomycetes* | 0.031 | *Latescibacterota* | 0.009 |
| *Chloroflexi* | 0.012 | *Planctomycetota* | 0.009 |

region was used for these analyses, the threshold for the minimum sequence length was lowered. The parameters used for this dataset can be found in Tables 5 and 6 in the Appendix. Comparison of the RiboSnake results with the original results proved difficult, as only a little information concerning the observed bacterial abundances was included in the publications. The comparison of the ten most abundant phyla for the switchgrass soil from the original study and the re-analysis with RiboSnake showed a high level of similarity when considering changes in nomenclature. The exceptions were the phyla *Chloroflexi*, *Actinobacteria* and the class *Betaproteobacteria*, which were found in the original study with a rather low abundance but were not found in the analysis with RiboSnake (Table 4). This could be the result of the adjusted abundance filter or the fact that not all samples from the original study were re-analyzed, but only a subset of eight samples. In conclusion, the analysis conducted with RiboSnake yielded consistent results that were comparable with those presented in the published research concerning the identified bacteria and their abundance. This evidence supports the applicability of RiboSnake for this type of data.

## CONCLUSION

RiboSnake provides a well-engineered, easy-to-install, and easy-to-use solution for *16S rRNA* analyses. We demonstrated that using verified parameter sets, the MOCK community composition was recovered and results obtained in previous publications could be attained. RiboSnake provides ready-to-use parameter sets for a range of scenarios and allows for the easy addition of new parameter sets. This, together with the all-in-one analysis, which provides users with a plethora of results and graphics generated by a fully automated pipeline, distinguishes RiboSnake as a tool with unique features.

We are convinced that by adding *in vitro* verified parameter sets and offering a mechanism for the community to extend those parameter sets, we are opening up an already well-established technique to a new group of users and assisting with reproducible science.

## AVAILABILITY OF SOURCE CODE AND REQUIREMENTS

- Project name: RiboSnake: 16S rRNA analysis workflow with QIIME2 and Snakemake
- Project home page: https://github.com/IKIM-Essen/RiboSnake
- Operating system(s): Linux
- Programming language: Python

- Other requirements: Conda/Mamba
- License: MIT license
- RRID:SCR_025608.

## DATA AVAILABILITY

The data sets supporting the results of this article and generated for this study are available in the NCBI repository under bioproject PRJEB75965. Snapshots of the code, results files and data access forms are available in GigaDB [61]. The workflow is hosted in workflowhub.eu [62].

## LIST OF ABBREVIATIONS

ASV, amplicon sequence variants; OTU, operational taxonomic units; PCoA, principal coordinate analysis.

## DECLARATIONS

### Ethical approval

The collection of samples from hand swabs as well as from the plasma of participants was approved by the Ethics Committee of the Medical Faculty of the University of Bonn (approval numbers 253/16 and 085/20). Consent for participation and publication was given by all participants.

### Competing interests

The authors declare no competing interests.

### Authors' contributions

AKD conceptualized the system, implemented the software, performed the analysis and wrote the original draft. JW, AD, JG, HM conducted the sequencing experiments. JW, AD and IK validated the pipeline. JW and AD helped write the draft manuscript. JK supervised part of the work and helped with the concept. RS helped with the ethics approvals and the generation of the samples. IK provided project administration. FM and IK provided supervision, conceptualization, methodology, and formal analysis, and they reviewed and edited the draft manuscript.

### Funding

We acknowledge funding from 01ZZ2013- SMITH - Medical Informatics Initiative Germany, PI: Folker Meyer. The funders played no role in the design or execution of the study. We used internal resources to cover the sequencing and laboratory costs incurred.

### Acknowledgements

We want to thank Prof. Dr. Johannes Köster for support with technical questions. We also thank the "Emschergenossenschaft", one of the two public utility companies in Essen (Wasserwirtschaftsverband) responsible for wastewater treatment, for their help with obtaining the wastewater samples used for testing the pipeline.

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

## APPENDIX

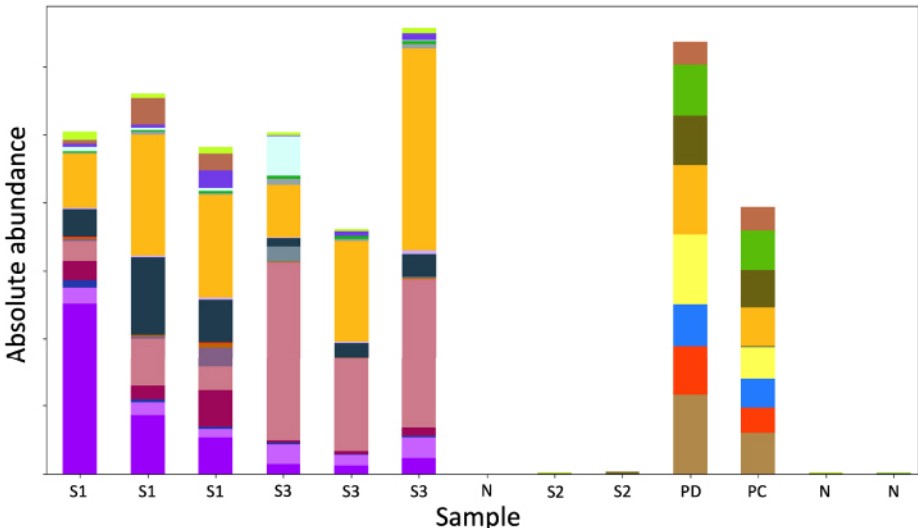

**Figure 5.** Absolute abundances for hands and surface swabs. Every color represents a bacterial genus. Legend can be found in Table 6 of the Appendix.

d__Bacteria; p__Actinobacteriota; c__Actinobacteria; o__Corynebacteriales; f__Corynebacteriaceae; g__Corynebacterium
d__Bacteria; p__Actinobacteriota; c__Actinobacteria; o__Corynebacteriales; f__Corynebacteriaceae; g__Lawsonella
d__Bacteria; p__Actinobacteriota; c__Actinobacteria; o__Micrococcales; f__Micrococcaceae; g__Micrococcus
d__Bacteria; p__Actinobacteriota; c__Actinobacteria; o__Micrococcales; f__Micrococcaceae; g__Rothia
d__Bacteria; p__Actinobacteriota; c__Actinobacteria; o__Propionibacteriales; f__Propionibacteriaceae; g__Cutibacterium
d__Bacteria; p__Bacteroidota; c__Bacteroidia; o__Flavobacteriales; f__Weeksellaceae; g__Chryseobacterium
d__Bacteria; p__Firmicutes; c__Bacilli; o__Bacillales; f__Bacillaceae; g__Bacillus
d__Bacteria; p__Firmicutes; c__Bacilli; o__Lactobacillales; f__Carnobacteriaceae; g__Granulicatella
d__Bacteria; p__Firmicutes; c__Bacilli; o__Lactobacillales; f__Enterococcaceae; g__Enterococcus
d__Bacteria; p__Firmicutes; c__Bacilli; o__Lactobacillales; f__Lactobacillaceae; g__Lactobacillus
d__Bacteria; p__Firmicutes; c__Bacilli; o__Lactobacillales; f__Listeriaceae; g__Listeria
d__Bacteria; p__Firmicutes; c__Bacilli; o__Lactobacillales; f__Streptococcaceae; g__Lactococcus
d__Bacteria; p__Firmicutes; c__Bacilli; o__Lactobacillales; f__Streptococcaceae; g__Streptococcus
d__Bacteria; p__Firmicutes; c__Bacilli; o__Staphylococcales; f__Gemellaceae; g__Gemella
d__Bacteria; p__Firmicutes; c__Bacilli; o__Staphylococcales; f__Staphylococcaceae; g__Staphylococcus
d__Bacteria; p__Firmicutes; c__Clostridia; o__Peptostreptococcales-Tissierellales; f__Peptostreptococcales-Tissierellales; g__Anaerococcus
d__Bacteria; p__Firmicutes; c__Clostridia; o__Peptostreptococcales-Tissierellales; f__Peptostreptococcales-Tissierellales; g__Finegoldia
d__Bacteria; p__Proteobacteria; c__Alphaproteobacteria; o__Rhodobacterales; f__Rhodobacteraceae; g__Paracoccus
d__Bacteria; p__Proteobacteria; c__Gammaproteobacteria; o__Burkholderiales; f__Neisseriaceae; g__Neisseria
d__Bacteria; p__Proteobacteria; c__Gammaproteobacteria; o__Burkholderiales; f__Neisseriaceae; g__uncultured
d__Bacteria; p__Proteobacteria; c__Gammaproteobacteria; o__Enterobacterales; f__Enterobacteriaceae; g__Escherichia-Shigella
d__Bacteria; p__Proteobacteria; c__Gammaproteobacteria; o__Enterobacterales; f__Enterobacteriaceae; g__Salmonella
d__Bacteria; p__Proteobacteria; c__Gammaproteobacteria; o__Pseudomonadales; f__Moraxellaceae; g__Enhydrobacter
d__Bacteria; p__Proteobacteria; c__Gammaproteobacteria; o__Pseudomonadales; f__Pseudomonadaceae; g__Pseudomonas

**Figure 6.** Legend for the absolute taxa barplot.

**Table 5.** Table listing parameters for human and environmental sampling.

| Parameter | Value human sampling | Value wastewater sampling | Value soil sampling |
|---|---|---|---|
| remove-columns | ["medication"] | ["name_abbreviation"] | ["site_name"] |
| **primertrimming** | | | |
| error_rate | 0.1 | 0.1 | 0.1 |
| rep_times | 2 | 2 | 2 |
| overlap | 5 | 5 | 5 |
| min_length | 8 | 8 | 8 |
| **sequence_joining** | | | |
| seq_join_length | 30 | 30 | 30 |
| minlen | 1 | 1 | 1 |
| maxdiffs | 10 | 10 | 10 |
| threads | 4 | 4 | 4 |
| threads | 15 | 15 | 15 |
| **filtering** | | | |
| relative-abundance-filter | 0.001 | 0.0005 | 0.0005 |
| min-seq-length | 200 | 180 | 140 |
| phred-score | 20 | 20 | 20 |
| max-ambiguity | 50 | 50 | 50 |
| min-length-frac | 0.75 | 0.75 | 0.75 |
| chimera-minh | 0.35 | 0.35 | 0.35 |
| perc-identity | 0.93 | 0.93 | 0.93 |
| perc-query-aligned | 0.93 | 0.93 | 0.93 |
| **metadata-parameters** | | | |
| taxa-heatmap-column | subject | name_abbreviation | extract_group_no |
| beta-metadata-column | subject | name_abbreviation | extract_group_no |
| gneiss-metadata-column | subject | name_abbreviation | site_name |
| cluster | features | features | features |
| **classification** | | | |
| perc-identity | 0.97 | 0.97 | 0.97 |
| maxaccepts | 1 | 1 | 1 |
| maxrejects | 1 | 1 | 1 |
| query-cov | 0.8 | 0.8 | 0.8 |
| min-consensus | 0.51 | 0.51 | 0.51 |
| **clustering** | | | |
| perc-identity | 0.99 | 0.99 | 0.99 |
| **rarefaction** | | | |
| max-depth | 500 | 500 | 500 |
| sampling_depth | 100 | 100 | 100 |
| repeats | 50 | 50 | 50 |
| metric | euclidean | euclidean | euclidean |
| clustering_method | nj | nj | nj |
| **diversity** | | | |
| **alpha** | | | |
| diversity-metric | ["pielou_e"] | ["pielou_e"] | ["pielou_e"] |
| phylogeny-metric | ["faith_pd"] | ["faith_pd"] | ["faith_pd"] |
| correlation-method | 'spearman' | 'spearman' | 'spearman' |
| **beta** | | | |
| diversity-metric | ["euclidean"] | ["euclidean"] | ["euclidean"] |
| diversity-pseudocount | 1 | 1 | 1 |
| diversity-n-jobs | 3 | 3 | 3 |
| correlation-method | 'spearman' | 'spearman' | 'spearman' |
| correlation-permutations | 999 | 999 | 999 |
| phylogeny-metric | weighted_normalized_unifrac | weighted_normalized_unifrac | weighted_normalized_unifrac |
| phylogeny-variance-adjusted | True | True | True |
| **dada2-paired** | | | |
| trunc-len-f | 240 | 240 | 240 |
| trunc-len-r | 240 | 240 | 240 |
| trim-left-f | 10 | 10 | 10 |
| trim-left-r | 10 | 10 | 10 |
| max-ee-f | 2.0 | 2.0 | 2.0 |
| max-ee-r | 2.0 | 2.0 | 2.0 |
| trunc-q | 5 | 5 | 5 |
| min-overlap | 12 | 12 | 12 |
| pooling-method | 'independent' | 'independent' | 'independent' |
| chimera-method | 'consensus' | 'consensus' | 'consensus' |
| min-fold-parent-over-abundance | 1.0 | 1.0 | 1.0 |
| n-reads-learn | 1,000,000 | 1,000,000 | 1,000,000 |

**Table 6.** Table listing parameters for human and environmental sampling.

| Parameter | Value human sampling | Value environmental sampling | Value soil sampling |
|---|---|---|---|
| **dada2-single** | | | |
| trunc-len | 240 | 240 | 240 |
| trim-left | 10 | 10 | 10 |
| max-ee | 2.0 | 2.0 | 2.0 |
| trunc-q | 5 | 5 | 5 |
| pooling-method | 'independent' | 'independent' | 'independent' |
| chimera-method | 'consensus' | 'consensus' | 'consensus' |
| min-fold-parent-over-abundance | 1.0 | 1.0 | 1.0 |
| n-reads-learn | 1,000,000 | 1,000,000 | 1,000,000 |
| **songbird** | | | |
| differential_prior | 0.5 | 0.5 | 0.5 |
| min_sample_count | 0 | 0 | 0 |
| min_feature_count | 0 | 0 | 0 |
| summary_interval | 1 | 1 | 1 |
| formula | "sex+age" | "sampling_date+sampling_site" | "extract_concen+elevation" |
| **ancom** | | | |
| metadata-column | ["subject"] | ["sampling_date","sampling_site"] | ["extract_group_no"] |
| **longitudinal-params** | | | |
| state_column | temperature | temperature | average_soil_temperature |
| individual_id_column | "name_abbreviation" | "name_abbreviation" | "name_abbreviation" |
| formula | "faith-pd+extraction_date" | "faith-pd+temperature" | "faith-pd+temperature" |

**Table 7.** List of the different tools and their usage.

| Tool | Method |
|---|---|
| QIIME2 [14] | General analysis methods |
| Snakemake [18] | Workflow management |
| FastQC [32] | Quality control |
| MultiQC [34] | Quality control visualization |
| Cutadapt [35] | Trimming |
| Kraken2 [33] | General overview |
| BLAST [37] | Filtering host contamination (default) |
| Bowtie2 [38] | Filtering host contamination (optional) |
| DADA2 [43] | ASV clustering |
| Ancom [49] | Feature importance analysis |
| gneiss [50] | Feature importance analysis |
| vsearch [36] | OTU clustering and classification |
| Songbird [51] | Feature importance analysis |
| qurro [52] | Visualization of songbird output |
| RESCRIPt [46] | Database creation from NCBI data |

■ d__Bacteria;p__Firmicutes;c__Bacilli;o__Staphylococcales;f__Staphylococcaceae;g__Staphylococcus

■ d__Bacteria;p__Actinobacteriota;c__Actinobacteria;o__Propionibacteriales;f__Propionibacteriaceae;g__Cutibacterium

■ d__Bacteria;p__Actinobacteriota;c__Actinobacteria;o__Corynebacteriales;f__Corynebacteriaceae;g__Corynebacterium

■ d__Bacteria;p__Firmicutes;c__Bacilli;o__Lactobacillales;f__Streptococcaceae;g__Streptococcus

■ d__Bacteria;p__Firmicutes;c__Bacilli;o__Bacillales;f__Bacillaceae;g__Bacillus

■ d__Bacteria;p__Firmicutes;c__Bacilli;o__Lactobacillales;f__Listeriaceae;g__Listeria

■ d__Bacteria;p__Proteobacteria;c__Gammaproteobacteria;o__Enterobacterales;f__Enterobacteriaceae;g__Salmonella

■ d__Bacteria;p__Proteobacteria;c__Gammaproteobacteria;o__Enterobacterales;f__Enterobacteriaceae;g__Escherichia-Shigella

■ d__Bacteria;p__Actinobacteriota;c__Actinobacteria;o__Corynebacteriales;f__Corynebacteriaceae;g__Lawsonella

■ d__Bacteria;p__Actinobacteriota;c__Actinobacteria;o__Micrococcales;f__Micrococcaceae;g__Rothia

■ d__Bacteria;p__Firmicutes;c__Bacilli;o__Lactobacillales;f__Lactobacillaceae;g__Lactobacillus

■ d__Bacteria;p__Firmicutes;c__Bacilli;o__Lactobacillales;f__Enterococcaceae;g__Enterococcus

■ d__Bacteria;p__Proteobacteria;c__Alphaproteobacteria;o__Rhodobacterales;f__Rhodobacteraceae;g__Paracoccus

■ d__Bacteria;p__Proteobacteria;c__Gammaproteobacteria;o__Burkholderiales;f__Neisseriaceae;g__uncultured

■ d__Bacteria;p__Proteobacteria;c__Gammaproteobacteria;o__Pseudomonadales;f__Pseudomonadaceae;g__Pseudomonas

■ d__Bacteria;p__Proteobacteria;c__Gammaproteobacteria;o__Burkholderiales;f__Neisseriaceae;g__Neisseria

■ d__Bacteria;p__Proteobacteria;c__Gammaproteobacteria;o__Pseudomonadales;f__Moraxellaceae;g__Enhydrobacter

■ d__Bacteria;p__Bacteroidota;c__Bacteroidia;o__Flavobacteriales;f__Weeksellaceae;g__Chryseobacterium

■ d__Bacteria;p__Actinobacteriota;c__Actinobacteria;o__Micrococcales;f__Micrococcaceae;g__Micrococcus

■ d__Bacteria;p__Firmicutes;c__Clostridia;o__Peptostreptococcales-Tissierellales;f__Peptostreptococcales-Tissierellales;g__Anaerococcus

■ d__Bacteria;p__Firmicutes;c__Bacilli;o__Lactobacillales;f__Streptococcaceae;g__Lactococcus

■ d__Bacteria;p__Firmicutes;c__Clostridia;o__Peptostreptococcales-Tissierellales;f__Peptostreptococcales-Tissierellales;g__Finegoldia

■ d__Bacteria;p__Proteobacteria;c__Gammaproteobacteria;o__Pasteurellales;f__Pasteurellaceae;g__Haemophilus

■ d__Bacteria;p__Proteobacteria;c__Gammaproteobacteria;o__Burkholderiales;f__Burkholderiaceae;g__Lautropia

■ d__Bacteria;p__Actinobacteriota;c__Actinobacteria;o__Micrococcales;f__Micrococcaceae;g__Kocuria

■ d__Bacteria;p__Firmicutes;c__Bacilli;o__Staphylococcales;f__Gemellaceae;g__Gemella

■ d__Bacteria;p__Firmicutes;c__Bacilli;o__Lactobacillales;f__Carnobacteriaceae;g__Granulicatella

■ d__Bacteria;p__Actinobacteriota;c__Actinobacteria;o__Corynebacteriales;f__Corynebacteriaceae;g__Corynebacteriaceae

■ d__Bacteria;p__Deinococcota;c__Deinococci;o__Deinococcales;f__Deinococcaceae;g__Deinococcus

■ d__Bacteria;p__Firmicutes;c__Clostridia;o__Peptostreptococcales-Tissierellales;f__Peptostreptococcales-Tissierellales;g__Peptoniphilus

■ d__Bacteria;p__Proteobacteria;c__Gammaproteobacteria;o__Enterobacterales;f__Enterobacteriaceae;g__Klebsiella

■ d__Bacteria;p__Firmicutes;c__Bacilli;o__Lactobacillales;f__Aerococcaceae;g__Abiotrophia

■ d__Bacteria;p__Proteobacteria;c__Gammaproteobacteria;o__Pseudomonadales;f__Moraxellaceae;g__Acinetobacter

■ d__Bacteria;p__Proteobacteria;c__Alphaproteobacteria;o__Rhodobacterales;f__Rhodobacteraceae;g__HIMB11

■ d__Bacteria;p__Proteobacteria;c__Alphaproteobacteria;o__Acetobacterales;f__Acetobacteraceae;g__Roseomonas

■ d__Bacteria;p__Firmicutes;c__Bacilli;o__Bacillales;f__Bacillaceae;g__Anaerobacillus

**Figure 7.** Legend for hand and keyboard swabs.



| | |
|---|---|
| 🟩 | d__Bacteria;p__Firmicutes;c__Bacilli;o__Staphylococcales;f__Staphylococcaceae;g__Staphylococcus |
| 🟪 | d__Bacteria;p__Actinobacteriota;c__Actinobacteria;o__Propionibacteriales;f__Propionibacteriaceae;g__Cutibacterium |
| 🟧 | d__Bacteria;p__Actinobacteriota;c__Actinobacteria;o__Corynebacteriales;f__Corynebacteriaceae;g__Corynebacterium |
| 🟨 | d__Bacteria;p__Firmicutes;c__Bacilli;o__Lactobacillales;f__Streptococcaceae;g__Streptococcus |
| 🟦 | d__Bacteria;p__Firmicutes;c__Bacilli;o__Bacillales;f__Bacillaceae;g__Bacillus |
| 🟥 | d__Bacteria;p__Proteobacteria;c__Gammaproteobacteria;o__Enterobacterales;f__Enterobacteriaceae;g__Escherichia-Shigella |
| 🟫 | d__Bacteria;p__Firmicutes;c__Bacilli;o__Lactobacillales;f__Listeriaceae;g__Listeria |
| ⬛ | d__Bacteria;p__Proteobacteria;c__Gammaproteobacteria;o__Enterobacterales;f__Enterobacteriaceae;g__Salmonella |
| 🟩 | d__Bacteria;p__Actinobacteriota;c__Actinobacteria;o__Corynebacteriales;f__Corynebacteriaceae;g__Lawsonella |
| 🟪 | d__Bacteria;p__Actinobacteriota;c__Actinobacteria;o__Micrococcales;f__Micrococcaceae;g__Rothia |
| 🟧 | d__Bacteria;p__Firmicutes;c__Bacilli;o__Lactobacillales;f__Enterococcaceae;g__Enterococcus |
| 🟨 | d__Bacteria;p__Firmicutes;c__Bacilli;o__Lactobacillales;f__Lactobacillaceae;g__Lactobacillus |
| 🟦 | d__Bacteria;p__Proteobacteria;c__Alphaproteobacteria;o__Rhodobacterales;f__Rhodobacteraceae;g__Paracoccus |
| 🟥 | d__Bacteria;p__Proteobacteria;c__Gammaproteobacteria;o__Burkholderiales;f__Neisseriaceae;g__uncultured |
| 🟫 | d__Bacteria;p__Proteobacteria;c__Gammaproteobacteria;o__Pseudomonadales;f__Pseudomonadaceae;g__Pseudomonas |
| ⬛ | d__Bacteria;p__Proteobacteria;c__Gammaproteobacteria;o__Pseudomonadales;f__Moraxellaceae;g__Enhydrobacter |
| 🟩 | d__Bacteria;p__Proteobacteria;c__Gammaproteobacteria;o__Burkholderiales;f__Neisseriaceae;g__Neisseria |
| 🟪 | d__Bacteria;p__Bacteroidota;c__Bacteroidia;o__Flavobacteriales;f__Weeksellaceae;g__Chryseobacterium |
| 🟧 | d__Bacteria;p__Actinobacteriota;c__Actinobacteria;o__Micrococcales;f__Micrococcaceae;g__Micrococcus |
| 🟨 | d__Bacteria;p__Firmicutes;c__Clostridia;o__Peptostreptococcales-Tissierellales;f__Peptostreptococcales-Tissierellales;g__Anaerococcus |
| 🟦 | d__Bacteria;p__Firmicutes;c__Bacilli;o__Lactobacillales;f__Streptococcaceae;g__Lactococcus |
| 🟥 | d__Bacteria;p__Firmicutes;c__Clostridia;o__Peptostreptococcales-Tissierellales;f__Peptostreptococcales-Tissierellales;g__Finegoldia |
| 🟫 | d__Bacteria;p__Firmicutes;c__Bacilli;o__Staphylococcales;f__Gemellaceae;g__Gemella |
| ⬛ | d__Bacteria;p__Proteobacteria;c__Gammaproteobacteria;o__Burkholderiales;f__Burkholderiaceae;g__Lautropia |
| 🟩 | d__Bacteria;p__Firmicutes;c__Bacilli;o__Lactobacillales;f__Carnobacteriaceae;g__Granulicatella |
| 🟪 | d__Bacteria;p__Proteobacteria;c__Gammaproteobacteria;o__Pasteurellales;f__Pasteurellaceae;g__Haemophilus |
| 🟧 | d__Bacteria;p__Firmicutes;c__Bacilli;o__Lactobacillales;f__P5D1-392;g__P5D1-392 |
| 🟨 | d__Bacteria;p__Actinobacteriota;c__Actinobacteria;o__Micrococcales;f__Micrococcaceae;g__Kocuria |
| 🟦 | d__Bacteria;p__Deinococcota;c__Deinococci;o__Deinococcales;f__Deinococcaceae;g__Deinococcus |
| 🟥 | d__Bacteria;p__Proteobacteria;c__Gammaproteobacteria;o__Enterobacterales;f__Enterobacteriaceae;g__Klebsiella |
| 🟫 | d__Bacteria;p__Firmicutes;c__Clostridia;o__Peptostreptococcales-Tissierellales;f__Peptostreptococcales-Tissierellales;g__Peptoniphilus |
| ⬛ | d__Bacteria;p__Firmicutes;c__Bacilli;o__Lactobacillales;f__Aerococcaceae;g__Abiotrophia |
| 🟩 | d__Bacteria;p__Proteobacteria;c__Gammaproteobacteria;o__Pseudomonadales;f__Moraxellaceae;g__Acinetobacter |
| 🟪 | d__Bacteria;p__Proteobacteria;c__Alphaproteobacteria;o__Acetobacterales;f__Acetobacteraceae;g__Roseomonas |
| 🟧 | d__Bacteria;p__Proteobacteria;c__Gammaproteobacteria;o__Xanthomonadales;f__Rhodanobacteraceae;g__Dokdonella |
| 🟨 | d__Bacteria;p__Proteobacteria;c__Alphaproteobacteria;o__Caulobacterales;f__Caulobacteraceae;g__Brevundimonas |

**Figure 8.** Legend for the comparison of the results with DADA2 and vsearch.



