## [Reviewer Report]

Indicate in the comments box below whether you are happy with the changes made or if the manuscript is unacceptable.Comments on revised manuscriptThe author's response has been fully addressed my concerns. The quality of the paper has apparently improved. I agree with the publication of this article.Indicate in the comments box below whether you are happy with the changes made or if the manuscript is unacceptable.Comments on revised manuscriptThe author's response has been fully addressed my concerns. The quality of the paper has apparently improved. I agree with the publication of this article.

---

## [Editor Report]

Editor’s AssessmentThis new software paper presents RiboSnake, a validated, automated, reproducible analysis pipeline implemented in the popular Snakemake workflow management system for microbiome analysis. Analysing16S rRNA gene amplicon sequencing data, this uses the widely used oQIIME2 [ tool as the basis of the workflow as it offers a wide range of functionality. Users of QIIME2 can be overwhelmed by the number of options at their disposal, and this workflow provides a fully automated and fully reproducible pipeline that can be easily installed and maintained. Providing an easy-to-navigate output accessible to non bioinformatics experts, alongside sets of already validated parameters for different types of samples. Reviewers requested some clarification for testing, comparisons and documentation, and this was improved to produce a convincingly easy-to-use workflow. Hopefully opening up an already very established technique to a new group of users and assisting them with reproducible science.This new software paper presents RiboSnake, a validated, automated, reproducible analysis pipeline implemented in the popular Snakemake workflow management system for microbiome analysis. Analysing16S rRNA gene amplicon sequencing data, this uses the widely used oQIIME2 [ tool as the basis of the workflow as it offers a wide range of functionality. Users of QIIME2 can be overwhelmed by the number of options at their disposal, and this workflow provides a fully automated and fully reproducible pipeline that can be easily installed and maintained. Providing an easy-to-navigate output accessible to non bioinformatics experts, alongside sets of already validated parameters for different types of samples. Reviewers requested some clarification for testing, comparisons and documentation, and this was improved to produce a convincingly easy-to-use workflow. Hopefully opening up an already very established technique to a new group of users and assisting them with reproducible science.Editor’s AssessmentThis new software paper presents RiboSnake, a validated, automated, reproducible analysis pipeline implemented in the popular Snakemake workflow management system for microbiome analysis. Analysing16S rRNA gene amplicon sequencing data, this uses the widely used oQIIME2 [ tool as the basis of the workflow as it offers a wide range of functionality. Users of QIIME2 can be overwhelmed by the number of options at their disposal, and this workflow provides a fully automated and fully reproducible pipeline that can be easily installed and maintained. Providing an easy-to-navigate output accessible to non bioinformatics experts, alongside sets of already validated parameters for different types of samples. Reviewers requested some clarification for testing, comparisons and documentation, and this was improved to produce a convincingly easy-to-use workflow. Hopefully opening up an already very established technique to a new group of users and assisting them with reproducible science.This new software paper presents RiboSnake, a validated, automated, reproducible analysis pipeline implemented in the popular Snakemake workflow management system for microbiome analysis. Analysing16S rRNA gene amplicon sequencing data, this uses the widely used oQIIME2 [ tool as the basis of the workflow as it offers a wide range of functionality. Users of QIIME2 can be overwhelmed by the number of options at their disposal, and this workflow provides a fully automated and fully reproducible pipeline that can be easily installed and maintained. Providing an easy-to-navigate output accessible to non bioinformatics experts, alongside sets of already validated parameters for different types of samples. Reviewers requested some clarification for testing, comparisons and documentation, and this was improved to produce a convincingly easy-to-use workflow. Hopefully opening up an already very established technique to a new group of users and assisting them with reproducible science.

---

## [Reviewer Report]

Reviewer name and names of any other individual's who aided in reviewerMichael HallDo you understand and agree to our policy of having open and named reviews, and having your review included with the published manuscript. (If no, please inform the editor that you cannot review this manuscript.)YesIs the language of sufficient quality?YesPlease add additional comments on language quality to clarify if neededIs there a clear statement of need explaining what problems the software is designed to solve and who the target audience is? YesAdditional CommentsIs the source code available, and has an appropriate Open Source Initiative license <a href="https://opensource.org/licenses" target="_blank">(https://opensource.org/licenses)</a> been assigned to the code?YesAdditional CommentsAs Open Source Software are there guidelines on how to contribute, report issues or seek support on the code?YesAdditional CommentsIs the code executable?YesAdditional CommentsIs installation/deployment sufficiently outlined in the paper and documentation, and does it proceed as outlined?Unable to testAdditional CommentsThe README states "If you want to test the RiboSnake functions yourself, you can use the same data used for the CI/CD tests." A worked example of how I can do this would be appreciated so I can test the workflow.Is the documentation provided clear and user friendly?YesAdditional CommentsIs there enough clear information in the documentation to install, run and test this tool, including information on where to seek help if required?NoAdditional CommentsThe Usage instructions say to create a new repository using ribosnake as a template, but ribosnake is not a template repository (see https://docs.github.com/en/repositories/creating-and-managing-repositories/creating-a-repository-from-a-template). The README states "If you want to test the RiboSnake functions yourself, you can use the same data used for the CI/CD tests." A worked example of how I can do this would be appreciated so I can test the workflow.Is there a clearly-stated list of dependencies, and is the core functionality of the software documented to a satisfactory level?YesAdditional CommentsHave any claims of performance been sufficiently tested and compared to other commonly-used packages? Not applicableAdditional CommentsIs test data available, either included with the submission or openly available via cited third party sources (e.g. accession numbers, data DOIs)?YesAdditional CommentsAre there (ideally real world) examples demonstrating use of the software? YesAdditional CommentsIs automated testing used or are there manual steps described so that the functionality of the software can be verified?YesAdditional CommentsYes, though as mentioned above, the README states "If you want to test the RiboSnake functions yourself, you can use the same data used for the CI/CD tests." A worked example of how I can do this would be appreciated so I can test the workflow.Any Additional Overall Comments to the AuthorThe Introduction could be make far more concise, there's a lot of repetition. The installation command in figure 1 is three commands, not two as stated in the text (third-last paragraph Introduction), and is slightly misleading from an installation point of view as it assumes conda and snakemake are installed. Though it is mentioned later in the text (p5) that snakemake and conda require manual installation. The in-text citation for Greengenes2 is just [?] - maybe a latex issue? The last paragraph of the 'Features and Implementations' section was mostly already stated earlier in the manuscript. Make the colouring consistent between fig 2a-c and 2d as well as the vertical ordering to make for easier comparison. For example, in figures 2a-c Enterococcus (grey) is on the bottom, whereas in fig 2d it is red and in the middle. Colour legends should also be added to Figures 3-5 to match Fig 2. A small table should be added showing the comparison of RiboSnake and the original publication for the top 10 most abundant phyla for the Atacama soil dataset and their abundances (see last paragraph of 'Usage and Findings'.RecommendationMinor Revisions

---

## [Reviewer Report]

Upload additional filesTRR-202405-01R01/stage_files/TRR-202405-01/Review MS/gx-TR-1716979819-revised.pdfReviewer name and names of any other individual's who aided in reviewerSalsabeel YousufDo you understand and agree to our policy of having open and named reviews, and having your review included with the published manuscript. (If no, please inform the editor that you cannot review this manuscript.)YesIs the language of sufficient quality?YesPlease add additional comments on language quality to clarify if neededIs there a clear statement of need explaining what problems the software is designed to solve and who the target audience is? YesAdditional CommentsIs the source code available, and has an appropriate Open Source Initiative license <a href="https://opensource.org/licenses" target="_blank">(https://opensource.org/licenses)</a> been assigned to the code?YesAdditional CommentsAs Open Source Software are there guidelines on how to contribute, report issues or seek support on the code?YesAdditional CommentsIs the code executable?YesAdditional CommentsIs installation/deployment sufficiently outlined in the paper and documentation, and does it proceed as outlined?YesAdditional CommentsIs the documentation provided clear and user friendly?YesAdditional CommentsIs there enough clear information in the documentation to install, run and test this tool, including information on where to seek help if required?YesAdditional CommentsIs there a clearly-stated list of dependencies, and is the core functionality of the software documented to a satisfactory level?YesAdditional CommentsHave any claims of performance been sufficiently tested and compared to other commonly-used packages? YesAdditional CommentsIs test data available, either included with the submission or openly available via cited third party sources (e.g. accession numbers, data DOIs)?YesAdditional CommentsAre there (ideally real world) examples demonstrating use of the software? YesAdditional CommentsIs automated testing used or are there manual steps described so that the functionality of the software can be verified?YesAdditional CommentsAny Additional Overall Comments to the AuthorThe manuscript presented by the authors describes a comprehensive study on the “RiboSnake pipeline” for 16S rRNA gene microbiome analysis, which is a user-friendly, robust, and multipurpose. RiboSnake, a validated, automated, reproducible QIIME2-based analysis pipeline implemented in Snakemake, offers parallel processing for efficient analysis of large datasets in both environmental and medical research contexts. Further demonstrating its effectiveness, this pipeline effectively analyzes human-associated microbiomes and environmental samples like wastewater and soil, thus expanding the scope of analysis for 16S rRNA data. The overall computational pipeline is useful and results are sound, validated through rigorous testing on MOCK communities and real-world datasets. However, there are some issues for improvement in the manuscript. Major comments: 1． In the clinical data section the author mentions rectal swabs were used from a published study [31]. While the source is referenced, it would be helpful to know if any information was provided in the referenced study regarding the collection methods or storage conditions for the rectal swabs. 2． The text mentions using cotton swabs pre-moistened with TE buffer + 0.5% Tween 20. While cotton swabs are common, are there any considerations for using different swab materials depending on the target analytes or sampling surface (e.g., flocked swabs for better epithelial cell collection)? 3． Does RiboSnake require user intervention during any steps, or is it fully automated? 4． The author mentions that contamination filtering parameters should be adjusted based on the sample type. How can users determine the appropriate filtering parameters for their specific samples? Are there guidelines for users to know how much adjustment is needed for specific scenarios? 5． The default abundance threshold for filtering low-frequency reads is chosen based on Nearing et al. [44]. Please discuss the rationale behind using a single threshold for all sample types? Would it be beneficial to allow users to define this threshold based on their data characteristics? 6． Would you like to explain the limitation of RiboSnake, such as specific types of samples it may not be suitable for or potential biases introduced by certain functionalities? 7． The manuscript mentions various visualization tools used throughout the pipeline (QIIME2, qurro). Please clarify which types of data are visualized with each tool, and how users can access or customize these visualizations? 8． To strengthen the manuscript's impact, consider discussing the specific novelty of RiboSnake compared to existing 16S rRNA gene microbiome analysis pipelines. Would you be able to elaborate on the unique features or functionalities of RiboSnake that address limitations of current methods? 9． EasyAmplicon is recently published pipeline and easy using in windows, mac and linux system, Minor comments: 1. Reference is missing in this sentence. “The default is the SILVA database [47]. Greengenes2 [? ] can be used alternatively”. 2. The author should careful about the lowercase and upper case throughout the manuscript. Please check the following for references:
..the 2017 published Atacama Soil data set with samples taken fromthe Atacama desert was used [32] as well as samples collected fromsoil under switchgrass published in [33]. based on an Euclidean beta diversitymetric, shows that the positive controls, as well as the samples taken from subjects 1 and 3 (S1 and S3), cluster together. A wide range of diversity analysis parameters are available in QIIME2 and its associated tools. These include the Shannon diversity index to measure richness, the Pielou index tomeasure evenness, or perform standard correlation analysis using Pearson or Spearman indices, among others. 3. In the introduction part this sentences “However, while these methods enable 16S rRNA analysis with minimal user interaction…” needs attention for clarity. Consider separating it into two sentences to emphasize the limitations of existing pipelines compared to the described methods’. Alternatively, using contrasting words like "in contrast" could highlight these differences. 4. More detail in attach PDF.RecommendationMajor Revisions